# Deep Brain Stimulation of the Basolateral Amygdala: Targeting Technique and Electrodiagnostic Findings

**DOI:** 10.3390/brainsci6030028

**Published:** 2016-08-10

**Authors:** Jean-Philippe Langevin, James W. Y. Chen, Ralph J. Koek, David L. Sultzer, Mark A. Mandelkern, Holly N. Schwartz, Scott E. Krahl

**Affiliations:** 1Neurosurgery Service, VA Greater Los Angeles Healthcare System, Los Angeles, CA 90073, USA; 2Neurology Service, VA Greater Los Angeles Healthcare System, Los Angeles, CA 90073, USA; james.chen1@va.gov; 3Psychiatry and Mental Health Service, VA Greater Los Angeles Healthcare System, Los Angeles, CA 90073, USA; ralph.koek@va.gov (R.J.K.); david.sultzer@va.gov (D.L.S.); holly.schwartz@va.gov (H.N.S.); 4Radiology Service, Nuclear Medicine VA Greater Los Angeles Healthcare System, Los Angeles, CA 90073, USA; mark.mandelkern@va.gov; 5Research and Development Service, VA Greater Los Angeles Healthcare System, Los Angeles, CA 90073, USA; scott.krahl@va.gov

**Keywords:** amygdala, basolateral nucleus, deep brain stimulation, microelectrode recording, PTSD, targeting technique

## Abstract

The amygdala plays a critical role in emotion regulation. It could prove to be an effective neuromodulation target in the treatment of psychiatric conditions characterized by failure of extinction. We aim to describe our targeting technique, and intra-operative and post-operative electrodiagnostic findings associated with the placement of deep brain stimulation (DBS) electrodes in the amygdala. We used a transfrontal approach to implant DBS electrodes in the basolateral nucleus of the amygdala (BLn) of a patient suffering from severe post-traumatic stress disorder. We used microelectrode recording (MER) and awake intra-operative neurostimulation to assist with the placement. Post-operatively, the patient underwent monthly surveillance electroencephalograms (EEG). MER predicted the trajectory of the electrode through the amygdala. The right BLn showed a higher spike frequency than the left BLn. Intra-operative neurostimulation of the BLn elicited pleasant memories. The monthly EEG showed the presence of more sleep patterns over time with DBS. BLn DBS electrodes can be placed using a transfrontal approach. MER can predict the trajectory of the electrode in the amygdala and it may reflect the BLn neuronal activity underlying post-traumatic stress disorder PTSD. The EEG findings may underscore the reduction in anxiety.

## 1. Introduction

The amygdala is a critical node within a network that regulates emotions. Two functional zones of the amygdala are especially important in this role. The centromedial group is the output of the amygdala: through its efferent connections, it orchestrates the physiological and behavioral components of emotions [1]. The basolateral group receives sensory input and it links emotions to stimuli [2,3]. This function is particularly important because it defines the behavior manifested in response to a stimulus. To fulfill its role, the basolateral amygdala receives modulatory inputs from the medial prefrontal cortex (mPFC) and several other regions, such as the hippocampus and the ventral tegmental area [4,5].

Because it defines the emotional context of behavior, the amygdala has been surgically targeted for intractable psychiatric conditions. Bilateral transfrontal stereotactic amygdalotomies have been performed for patients suffering from intractable aggressiveness [6]. In these cases, aggression was targeted as a symptom regardless of the specific underlying psychiatric condition. The amygdala was understood as a center of fear and anger. Recent studies have shown that the amygdala links both positive as well as negative emotions to stimuli (for review [3]). The amygdala determines if a stimulus should elicit fear (association) or not (extinction). In fact, the basolateral nucleus of the amygdala BLn contains “fear cells” that are active during fear acquisition and consolidation and “extinction cells” that are active during fear extinction [4]. Emotional extinction is a delicate process by which the basolateral nucleus (BLn) learns to no longer elicit an emotion in response to a stimulus. This learning process likely requires plasticity in the BLn, in part induced by modulation from the mPFC [2], whereby a larger population of extinction cells is activated relative to fear cells. In this sense, the BLn can be seen as an emotional receptive field modulated by the mPFC, the hippocampus and several other regions [7]. The relative population of active fear cells and extinction cells can therefore determine the specific response to a reminder. An intense stimulus, such as a life-threatening event, may overwhelm this learning process by forming a stereotypical neurophysiological response in the BLn. In post-traumatic stress disorder (PTSD), a life-threatening event leads to a state of failure of fear extinction that may be understood as a neuroplasticity failure to re-engage extinction cells. In this context, neuromodulation could be focally applied to the BLn to restore fear extinction by modifying the relative volume of active fear cells compared to extinction cells.

Indeed, we have previously shown that BLn deep brain stimulation (DBS) can promote fear extinction in a rodent model of PTSD [8,9]. This effect could translate in patients suffering from PTSD. We are performing an early-phase trial evaluating BLn DBS for treatment-refractory PTSD [10]. In this paper, we aim to describe our targeting technique and intraoperative microelectrode recording (MER) results in a case of bilateral BLn DBS electrode placements. We are aware of only one other human BLn DBS case: a teenage boy was treated successfully for treatment-refractory self-mutilating behavior in the context of severe autism [11].

## 2. Materials and Methods

### 2.1. Subject

Our subject is a Gulf War veteran who developed PTSD from participation in a military assault and the subsequent exposure to the bodies of enemy combatants. His baseline Clinician Administered PTSD Scale (CAPS) [12] score was 119, classifying him among the most severely ill patients. He suffered from vivid nightmares during which charred corpses would surround him. Upon exposure to trauma reminders, he would enter an unresponsive, hyper-aroused, dissociated state. He often missed workdays after a triggered flashback and lost several jobs due to absenteeism. He remained severely symptomatic despite 20 years of antidepressant, antipsychotic, antiadrenergic (prazosin) and mood stabilizer pharmacotherapy and cognitive-behavioral psychotherapy. The patient met all the criteria for enrollment in our clinical trial [10], and he gave his informed consent for inclusion before participating in the study. The study was conducted in accordance with the Declaration of Helsinki, and the protocol was approved by the institutional review board of the Greater Los Angeles VA Healthcare System (IRB approval code is PCC#2016-040351).

### 2.2. Targeting

The patient underwent a stereotactic 3T MRI with gadolinium. The images were re-oriented along the anterior commissure/posterior commissure (AC/PC) in our planning software (FHC, Bowdoin, ME, USA). Targeting the BLn is complicated due to anatomical variations in this region. Using a stereotactic atlas [13], the inferior limit of the BLn is located 16 mm lateral to the AC, 4 mm posterior to the AC and 18 mm inferior to the AC-PC plane. However, these coordinates would be incorrect for a large number of patients, and targeting has to be performed based on the local mesiotemporal anatomy.

The BLn is located in the inferior portion of the amygdala. It is at the center of the amygdala where it is flanked laterally by the lateral nucleus and medially by the basomedial and basolateral ventromedial part [14]. The central and medial nuclei of the amygdala are dorsal to the BLn. When the MRI is oriented along the AC-PC plane, the fornices can be seen crossing the hypothalamus in the same coronal plane as the BLn within the amygdala [14]. Within this coronal plane, the BLn is located in the center of the amygdala from a medial-to-lateral perspective. When studying the axial plane at this level, the BLn is located just anterior to the tip of the temporal horn, which thus serves as another important landmark [11]. Finally, the inferior border of the BLn is marked by the presence of the head of the hippocampus (Figure 1).

The electrode trajectory must avoid critical structures to permit a safe transfrontal approach. It should be posterior to the lenticulostriate perforator vessels from the first segment of the middle cerebral artery and it should remain lateral to the superior aspect of the ambient cistern (Figure 1). Finally, the trajectory should be seen traversing just anterior to the tip of the temporal horn to confirm the location within the BLn [11]. The location of the entry point will vary depending on the anatomy of the critical structures. In general, it will have a lateral angle of 0–10° from midline and an anterior angle of 70–80° from the AC-PC plane.

### 2.3. Anesthesia and Microelectrode Recording

Induction was performed with fentanyl and propofol. Subsequently, the patient received sevoflurane until the burr holes were completed. At that point, sevoflurane was stopped, the patient was awakened and the laryngeal mask was removed. The microelectrode recording (MER) was started on the right side, once the patient had fully regained consciousness, using a platinum-iridium differential electrode (DZAP, FHC). The patient received fentanyl (100 mcg) and midazolam (1 mg) during the right MER and he received fentanyl (50 mcg) and midazolam (1 mg) during the left MER. This medication was administered in order to assist the patient with symptoms of pain and anxiety that became clinically significant as the patient emerged from anesthesia.

The MER was started 20 mm above the target. The microelectrodes were advanced and the signal output was recorded for 30 s at 0.5 mm increments. The unit signals were sampled at 48,000 Hz and were recorded with low- and high-frequency cutoffs of 500 and 5000 Hz, respectively. The spike counts were calculated with a spike threshold set at 150 μvolt (MATLAB, MathWorks, Natick, MA, USA). The spike detection algorithm follows a few rules for automatic spike detection. It only detected spikes in unilateral direction (on the upward deflection) when the signal data point reaches above the threshold of 150 μvolt. It requires the signal data point to decrease below 150 μvolt before a new spike is counted. This algorithm biased toward a low estimate of the actual spike generated because in scenario that multiple discharging units were overlapping one after the other and the waveform stayed above the threshold, the multiple spikes were counted as the same spike. However, this algorithm could avoid a bigger bias by counting every data point above the threshold as a new spike. Spikes were counted per second over the 30 s period, and the mean and standard deviation (*n* = 30) of spike frequency was calculated per recording site.

### 2.4. EEG Recording

The EEG recording was performed using a clinical EEG machine (Nihon Kohden, Irvine, CA, USA). The electrodes were placed according to the standard international 10–20 system. The EEG tracings were displayed using a double-banana montage. The EEG was studied using conventional clinical analysis techniques.

### 2.5. Sleep and Nightmares Recording

The most distressing PTSD symptom reported by our patient was the occurrence of severe nightmares. Given the clinical relevance of those episodes for our subject, we elected to measure the impact of BLn DBS on sleep and nightmares. Upon enrollment, he was given a table on which he recorded the hours that he slept every night as well as the number and the subjective intensity (0 none, 10 most severe) of these nightmares. This method of measurement is inherently qualitative, subjective and non-validated. Our aim was to obtain a preliminary sense of the impact of our treatment on subjective sleep experience.

## 3. Results

### 3.1. Microelectrode and DBS Lead Positions

The positions of the DBS leads were confirmed using a stereotactic CT scan fused to the preoperative MRI using our planning software (WayPoint, FHC, Bowdoin, ME, USA). The location of microelectrode recordings in relation to the anatomy was extrapolated based on a stereotactic atlas [14], while taking into account the local anatomy and the AC-PC coordinates at each MER recording point.

### 3.2. Microelectrode Recording

The MER data correlated to the anatomy as predicted by a stereotactic atlas (Figure 2). The initial high-frequency signal was in the position of the ventral globus pallidus externa (GPe) (49 ± 17 Hz on the left and 114 ± 84 Hz on the right). This signal was followed by an area sparse in spikes in the region of the ventroamygdalofugal pathway dorsal to the amygdala. The entry into the amygdala was characterized by an increase in firing frequency in the region of the central nucleus of the amygdala CeA (2 ± 3 Hz on the left and 20 ± 18 Hz on the right). The BLn was characterized by a similar neuronal firing frequency as the CeA. The dorsal BLn showed 3 ± 3 Hz on the left and 29 ± 19 Hz on the right. The ventral BLn had a spike frequency of 12 ± 13 Hz and 33 ± 30 Hz on the left and right, respectively. The ventral exit from the BLn was characterized with a drop of spikes and background activity consistent with white matter in the region separating the BLn from the hippocampus. The entry into the hippocampus was characterized by high-frequency activity on the right (85 ± 53 Hz), but not on the left (2 ± 4 Hz). Overall, the frequency was higher in the right hemisphere compared to the left.

### 3.3. Intraoperative Neurostimulation

The patient was awake and able to correspond with the psychiatry team during testing of DBS electrode contacts. The intra-operative neurostimulation was conducted using the implant DBS electrode (3387, Medtronic, Minneapolis, MN, USA). The DBS electrodes were connected to an external neurostimulator (37022, Medtronic, USA) using an alligator clip wire (3550-67, Medtronic, USA) that allows for each contact to be activated individually. For each electrode contact, we used the following stimulation parameters: 120 μs of pulse width, 160 Hz of frequency and an amplitude of 0–5 V increased by increments of 0.5 V. The activation of the contacts located in the BLn triggered memories of places he had been, mainly in his childhood. Some of the scenes were experienced as if from a distance. These were generally experienced as pleasant or amusing. He also experienced emotions of calm. Since his diagnosis of PTSD, he had not had pleasant memories and this represented a new experience for him. The only unpleasant experience was a transient subjective sensation that he could not picture the examiner’s face when he closed his eyes, despite accurate recognition by looking at the examiner in person or on an identification card. There were no other adverse events related to the stimulation at amplitudes up to 5 V. Electrical stimulation of the mesiotemporal structures has led to similar experiences in epilepsy patients [15].

### 3.4. Post-Operative Electroencephalogram

The patient underwent surveillance of a 30 min EEG at baseline and then monthly post-operatively. The predominant muscle artifacts that were noted in awake recordings attenuated significantly after DBS stimulation. The persistent finding of being able to relax his frontalis muscles suggests that the patient was less anxious or hypervigilant. Over time, the EEG has demonstrated a reduction in the frequency of the posterior dominant rhythm (PDR) from 11 Hz to 9 Hz. Furthermore, the EEG showed progressively more sleep patterns and, by month 10, slow wave sleep was observed on all subsequent EEG studies (Figure 3). The patient reported improved duration and quality of sleep during the same interval. In particular, he reported an average gain of three hours of sleep and a marked reduction in nightmare frequency and severity (Figure 4). These nightmares were reported to the study team and many previous psychiatrists as horrifying re-experiences of combat events that woke him from sleep in an autonomically hyper-aroused state. For years he would get out of bed after these nightmares and stand in a cold shower for an hour or more, trying to “wash off” the residue of charred flesh he had experienced enveloping him during the nightmare. Upon enrollment in the study, given that this was the patient’s most distressing PTSD symptom, he was given a recording sheet on which he recorded the number and subjective (0 none, 10 most severe) intensity of these nightmares. As shown in Figure 4, these experiences have decreased in frequency and severity with DBS, and have not recurred as of the time of this writing (18 months after initiation of DBS).

## 4. Discussion

BLn DBS may prove beneficial for psychiatric conditions characterized by the failure to extinguish the link between an emotion (fear, pleasure) and a stimulus. Thus far, our patient has shown a substantial clinical improvement [16]. This is the second reported case of BLn DBS [11]. In both cases, a transfrontal trajectory was safely employed. This approach allows for a DBS lead placement that spans the CeA, the BLn, and the head of the hippocampus. These structures are interconnected and have been implicated in PTSD. The transfrontal approach therefore allows us to modulate more than one target nucleus depending on the clinical response. In addition, this trajectory avoids the ventricle and the body of the hippocampus, both of which would be entered during a transoccipital approach. Finally, the transfrontal approach is comfortable for the patient and it permits a safe and tolerable subdermal implantation of the leads. A transtemporal approach would require dividing the temporalis muscle, thus causing pain and possibly interfering with subcutaneous tunneling. The neurostimulation in this acute setting triggered memories of childhood; this experience has been reported with mesiotemporal electrical stimulation in epilepsy patients [15] and may prove predictive of proper electrode placement.

The MER findings were concordant with the anatomy as predicted by a stereotactic atlas. There is a relative paucity of neuronal spikes at the dorsal and the ventral extent of the BLn corresponding to white matter tracts at the margins of the nucleus. MER may predict the trajectory and span of the lead within the BLn. Our MER data also suggested more neuronal spikes in the right subcortical structures compared to the left. Several factors could account for this discrepancy. For instance, differences in anesthesia may account for this finding, although efforts were made to maintain the level of consciousness and the medication administered as constant during the right and the left MERs. Nevertheless, a benzodiazepine and a narcotic agent had to be administered to help the patient with pain and anxiety which became significant as the anesthesia was lifted. Other factors could also explain some of the differences between the right and left MER recordings. The trajectories may have been slightly different and the left electrode may have passed through a region with relative paucity in neurons compared to the right side. Alternatively, our MER findings may represent an overall increase in neuronal activity in the right hemisphere and the right amygdala in the context of PTSD. Other authors have reported hemispheric laterality findings in PTSD. Using resting-state magnetoenceopahlography (MEG), Engdahl et al. [17] showed that PTSD patients have more synchronous neuronal interactions in the right hemisphere compared to normal controls or PTSD patients in remission. Using a SPECT scan, Pagani et al. [18] showed an increase in right hemisphere cerebral blood flow when PTSD patients were exposed to a traumatic reminder compared to normal controls. Our ability to draw any conclusions in regard to this right and left discrepancy in MERs is limited by the fact that we have data for only one subject thus far.

The EEG findings of reduced PDR frequency and increased presence of sleep may reflect an effect of BLn DBS on anxiety. The PDR frequency has been reported to be higher in anxious patients compared to normal controls [19]. This effect on the PDR can serve as a proxy for anxiety whether it is directly related to anxiety or related to the patient’s inability to achieve a relaxed state [20]. Clinically, DBS of the amygdala reduced hyper-arousal symptoms and the underlying anxiety associated with PTSD. This effect may translate into an overall reduction in PDR and the appearance of slow-wave sleep SWS on the EEG since the patient is less anxious and is able to fall asleep easily. During the same interval, the patient reported a subjective improvement in duration and quality of sleep. By reducing anxiety, DBS may permit the activation of amygdala circuits critical to sleep. Anxiety can be understood as the psychological percept associated with the inability to activate those sleep circuits due to amygdala-mediated hypervigilance.

## 5. Conclusions

We describe our targeting technique and intraoperative findings for the placement of BLn DBS electrodes. BLn DBS may prove beneficial in conditions where emotional regulation mechanisms are dysfunctional, leading to a pervasive state of emotional inflexibility. In severe PTSD, failure of fear extinction causes a generalized association of fear to benign stimuli. In substance abuse, dysfunction in pleasure control leads to a preoccupation with drug-seeking behavior due to a failure of reward extinction. We describe MER findings for a patient with severe PTSD. The MER data may predict the span of the DBS lead within the BLn. Furthermore, the increase in spike frequency can be explained by the increase in metabolism and cerebral blood flow seen in PTSD patients. On EEG, chronic BLn DBS was associated with an overall reduction in the PDR frequency and the presence of SWS. These findings may highlight a reduction in anxiety. Nevertheless, our study is limited to a single subject. In addition, MER in the clinical setting does not clearly distinguish the type and the exact number of neurons being recorded, limiting our analytic capability.

## Figures and Tables

**Figure 1 brainsci-06-00028-f001:**
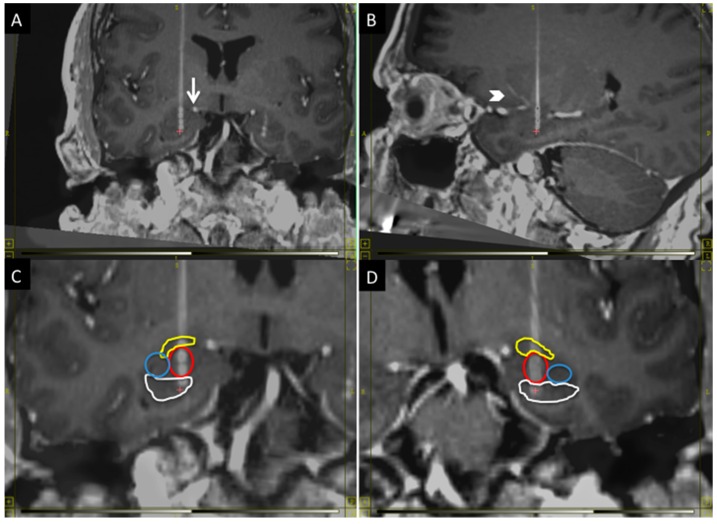
The figure shows the anatomical position of the BLn electrodes in the coronal plane (**A**,**C**,**D**) and sagittal plane (**B**): (**A**) The electrode must avoid the superior extent of the ambient cistern (arrow); (**B**) and the lenticulostriate vessels (arrowhead); (**C**,**D**) respectively show the right and left Bln electrodes with segmentation of the nuclei: in yellow, the central nucleus; in red, the basolateral nucleus; in blue, the lateral nucleus and, in white, the head of the hippocampus. The electrode contact distribution, from dorsal to ventral: Central nucleus (one contact); BLn (two contacts); Hippocampus (one contact). Laterally: Lateral nucleus (zero contacts).

**Figure 2 brainsci-06-00028-f002:**
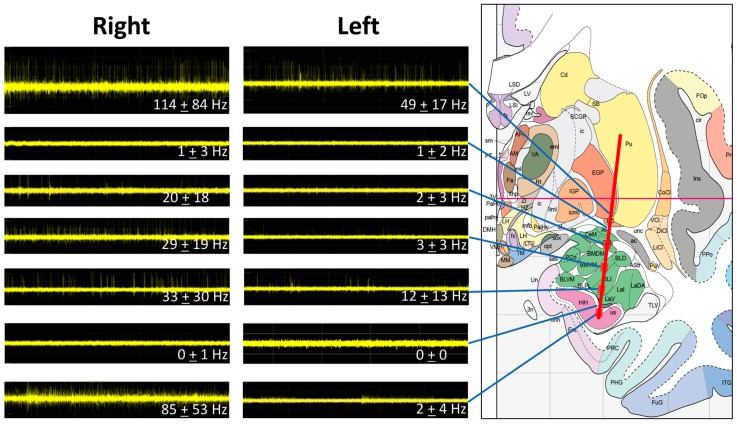
MER signals at different sites along the trajectory of the microelectrode. The predicted trajectory has been superimposed to a stereotactic brain atlas [14]. The MERs match the predicted anatomy with higher frequencies noted in the GPe, the amygdala and the hippocampus. The right side shows a higher frequency than the left side. The entrance and the exit from the amygdala are marked by a drop in background activity.

**Figure 3 brainsci-06-00028-f003:**
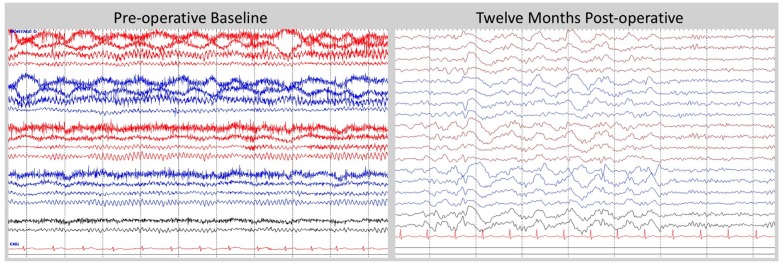
EEG at baseline and one year post-operatively. At baseline, the EEG shows robust PDR and muscle artifacts in the frontal channels, possibly related to anxiety. At one year post-operatively, the EEG shows a reduction in PDR and the presence of SWS, which may reflect a reduction in anxiety. Double Banana montage and a bandwidth of 1–70 Hz were used for displaying EEG tracings. Tracings from the right hemisphere were color-coded as red, and left hemisphere as blue, and sagittal line as black. The tracings from top to bottom in sequence were: F_p_2-F8, F8-T4, T4-T6, T6-O2, F_p_1-F7, F7-T3, T3-T5, T5-O1, F_p_2-F4, F4-C4, C4-P4, P4-O2, F_p_1-F3, F3-C3, C3-P3, P3-O1, Fz-Cz, Cz-Oz. One channel EKG tracing, which was in red color, was shown at the bottom.

**Figure 4 brainsci-06-00028-f004:**
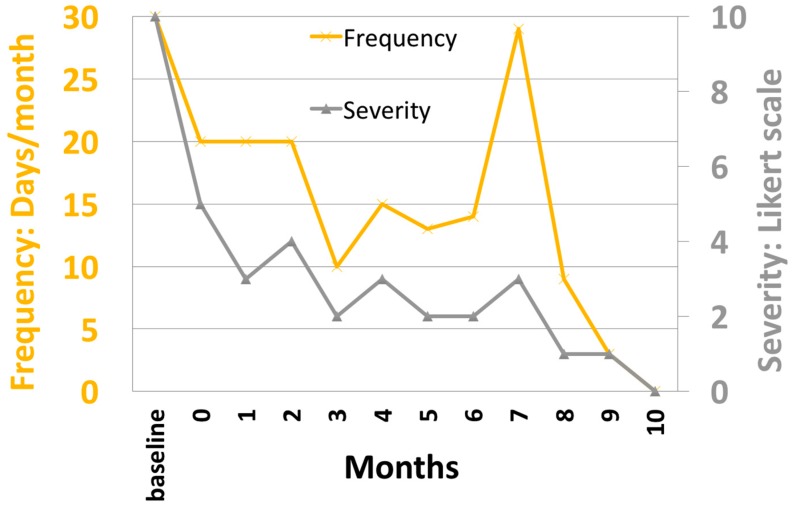
Graph showing the self-reported frequency and intensity of nightmares over time after BLn DBS. After month 10, the patient reported the occurrence of occasional bad dreams that were different from his typical nightmares in quality and intensity.

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
