# Peer review of "Deep Brain Stimulation of the Basolateral Amygdala: Targeting Technique and Electrodiagnostic Findings"

_brainsci, 2016, doi:10.3390/brainsci6030028_

Round 1
Reviewer 1 Report
In this article, the authors present their targeting technique and MER findings in a patient with PTSD treated with amygdala DBS.
Comments and questions:
- Line 42. La nucleus, Ce and BLn cannot be considered functional zones. They are nuclei. I would agree with terms such as Basolateral complex and corticomedial region. Those are broader. Nuclei are not zones.
- Lines 64-68. The BLn and basal ganglia structures have very different cellular and connection contents. Cannot assume based on results of DBS in the latter, that BLn stimulation will have an overall inhibitory effect.
- Why giving fentanyl and midazolan during the recordings and not trying to do it awake, as in PD?
- Recordings of striatal cells show fairly low firing cells. Though the ventral striatum/putamen is different, 49 and 114Hz are really high firing rates. Moreover, how can they justify such differences between hemispheres?
- Again, how can they explain 3vs 29Hz in amygdala firing in different hemispheres?
- 85 vs 2Hz. This difference is too striking. No biological mechanism I know could account for it. Was it a technical issue? Multiunit? Bursty cell patterns deviating the average?
- Can the authors describe the memories triggered by stimulation in further detail?
- How was severity of nightmare measured? Subjectively? Frequency also problematic as one often does not fully remember dreams. I would tone down this part of the description, as the authors do not have a proper way to quantitatively addressing the issue.
Reviewer 2 Report
1. the methods described here are too simple, the authors should describe more details, including:
(1) the data process of the recording?
(2) which type of electrode used for stimulation? and the parameters used for electric stimulation?
(3) the n value of recording sites? (all the standard deviations are too large compared with the mean values, for example, 12 ± 13 Hz and 33 ± 30 Hz)
(4) the recording sites of EEG? and the analysis method for EEG?
(5) How to perform the Likert scale?
2. Fig 1: the colored-circled areas should be explained.
3. Fig 2 : the words in the figure are too small to be recognized, and the traces of spikes are blurred.
4. Fig 3: the words in the figure are also too small to be recognized, and the legend is too brief.
5. Could the EEG index the level of anxiety? More references should be discussed.
Reviewer 3 Report
I would recommend that you for MER on future subjects with local anesthesia only for cleaner results.
Author Response
Thank you for your comment. We agree with you that using only local anesthesia would have provided better MER data. Unfortunately, was unable to tolerate the procedure due to his underlying severe PTSD. We was too anxious and would not have been able to complete the surgery. We added this explanation in section 2.3 Anesthesia and microelectrode recording.
Round 2
Reviewer 1 Report
The authors have satisfactorily addressed my comments. Still unclear how such huge differences in firing rate between hemispheres might have occurred. Yet, data is data. Please spell SWS.
Author Response
We would like to thank Reviewer 1 for his comments. As suggested, we spelled out "slow-wave sleep" at the first occurrence.
Reviewer 2 Report
minor:
The resolution of figure 3 is also blurred. And there were some green traces in it, however, not explained in legend.
Author Response
We would like to thank Reviewer 2 for his comments. We have improved figure 3 in this revision by improving the contrast and the brightness of the image. This should clarify the picture and optimize the quality. The EEG tracings are red (right hemisphere), blue (left hemisphere) or black (sagittal). The color green may have appeared as a result of poor contrast.